# Liquid-crystalline behavior on dumbbell-shaped colloids and the observation of chiral blue phases

Guangdong Chen[1,9], Hanwen Pei[2,9], Xuefei Zhang[3], Wei Shi[4], Mingjie Liu [4], Charl F. J. Faul [5], Bai Yang [1], Yan Zhao[6], Kun Liu [1,7], Zhongyuan Lu [1,8] ✉, Zhihong Nie [3] ✉ & Yang Yang [1] ✉

Colloidal liquid crystals are an emerging class of soft materials that naturally combine the unique properties of both liquid crystal molecules and colloidal particles. Chiral liquid crystal blue phases are attractive for use in fast optical displays and electrooptical devices, but the construction of blue phases is limited to a few chiral building blocks and the formation of blue phases from achiral ones is often counterintuitive. Herein we demonstrate that achiral dumbbell-shaped colloids can assemble into a rich variety of characteristic liquid crystal phases, including nematic phases with lock structures, smectic phase, and particularly experimental observation of blue phase III with double-twisted chiral columns. Phase diagrams from experiments and simulations show that the existence and stable regions of different liquid crystal phases are strongly dependent on the geometrical parameters of dumbbell-shaped colloids. This work paves a new route to the design and construction of blue phases for photonic applications.

Liquid crystals (LCs) are a state of matter that bears liquid-like fluidity and crystal-like ordering, which are ubiquitous in the natural world and our everyday life[1]. Blue phase (BP) LC is one of the most fascinating chiral LCs for applications in fast light modulators or tunable photonic crystals[2,3]. BPs with a network of double-twisted columns exist in a small temperature range between the isotropic and chiral nematic phases. There are three types of BPs, namely, BP I, BP II, and BP III. BP I and BP II possess body-centered and simple cubic structures, respectively, whereas BP III have amorphous structure[4,5]. Despite reasonable progress in the field, several barriers remain in the applications of BPs: (i) BPs only exist in a very narrow temperature range (usually less than a few kelvin)[6,7], (ii) only a few chiral

building blocks can form BPs, and (iii) the molecular origin for the chirality of BPs is still unclear[2,3]. Compared with molecular LCs, colloid-based LC materials are thermally stable, inexpensive, and possess enhanced susceptibility to external fields, including shear fields[8,9], electric fields, and magnetic fields[10–13], which makes them attractive for a wide range of applications such as shearing microlithography[9], and photonics[14,15]. Moreover, because of their unique dimensions (from submicrometer to micrometer scale), colloidal particles are ideal model systems for studying LC phase behavior of their molecular counterparts which are difficult to be directly observed and studied at single molecule level[16–22], to allow one to deduce a link between microscopic and macroscopic

[1]State Key Laboratory of Supramolecular Structure and Materials, College of Chemistry, Jilin University, Changchun 130012, China. [2]State Key Laboratory of Polymer Physics and Chemistry, Changchun Institute of Applied Chemistry, Chinese Academy of Sciences, Changchun 130022, China. [3]State Key Laboratory of Molecular Engineering of Polymers, Department of Macromolecular Science, Fudan University, Shanghai 200438, China. [4]Key Laboratory of Bioinspired Smart Interfacial Science and Technology of Ministry of Education, School of Chemistry, Beihang University, Beijing 100191, China. [5]School of Chemistry, University of Bristol, Bristol BS8 1TS, UK. [6]Jihua Institute of Biomedical Engineering and Technology, Jihua Laboratory, Foshan 528000, China. [7]State Key Laboratory of Applied Optics, Changchun Institute of Optics, Fine Mechanics and Physics, Chinese Academy of Sciences, Changchun 130012, China. [8]Institute of Theoretical Chemistry, College of Chemistry, Jilin University, Changchun 130023, China. [9]These authors contributed equally: Guangdong Chen, Hanwen Pei. ✉e-mail: luzhy@jlu.edu.cn; znie@fudan.edu.cn; yangchem@jlu.edu.cn

properties[23–26]. Therefore, great efforts are needed to develop colloidal BP LCs for the direct observation of BP.

Dumbbell-shaped colloids (DBCs) are featured with distinctive concave cavities between the spheroidal tips and unique packing behavior, making them attractive as building blocks for functional optical materials[27–31]. Nanosized dumbbells with charge patches can pack into zippers, cross-stacks, and open-lattice crystals[27]. Dumbbell-like nano-arrows can assemble into net-like, zipper-like, and weave-like two-dimensional lattices, as well as non-close-packed three-dimensional supercrystals but not in LC state, depending on the aspect ratio of the colloids[28]. Moreover, dumbbell-shaped molecules exhibit geometry-dependent assembly behavior to produce helical structures[32–35], suggesting the concave geometry can twist the packing of dumbbell-shaped building blocks. Colloidal LCs formed by DBCs would enable the visualization of the chiral/BPs with twisted/double-twisted structures at the single particle level, thus providing new insights into how chirality can emerge from achiral building blocks[36–38]. Nevertheless, to date, the manner in which the concave geometry of DBCs determines their LC phase behavior has not been explored experimentally, largely due to current challenges in the large-scale synthesis of these colloids with precisely controlled geometries.

This work presents a systematic study of the LC phase behavior of silica DBCs, which resemble dumbbell-shaped LC mesogens, with precisely tunable geometry. We observe that these silica DBCs assemble into a rich variety of characteristic LC phases, including double-twisted BP III, nematic phases with one-lock (N1) and two-lock (N2) structures, and smectic A (SmA) phase (Fig. 1). The phase behaviors of DBCs are found to be dictated by the two key geometrical parameters of DBCs, namely, the ratio of diameters of end blocks to central blocks ($R_D$), and the ratio of lengths of end blocks to central blocks ($R_L$), see schematic Fig. 1. Most importantly, this work demonstrate the first experimental observation of colloidal chiral BP III. In particular, the colloidal chiral BP III was formed from achiral building blocks in the range of $R_D > 1.1$ and $R_L < 0.45$. Brownian dynamic simulations confirm the geometry-dependent formation of mesophases and show good agreement with experimental results. The experimental and computational results are summarized in phase diagrams of stable phases and phase transitions in the $R_D$ and $R_L$ space. Moreover, we also demonstrate that the colloidal LC phases can be dynamically manipulated using magnetic fields. This study provides fundamental insights into the phase behavior of dumbbell-shaped objects, and the fabrication of BP LC materials for photonic applications.

## Results

### LC phases assembled from DBCs with various geometries

The silica DBCs with controlled $R_D$ (1.06–1.77), $R_L$ (0.08–1.81), and $L/D_e$ (1.55–9.93, where $L$ is the overall length of the DBC, i.e., $L = 2L_e + L_c$) were synthesized via a wet-chemical method[30,39]. The DBCs were dispersed in dimethyl sulfoxide (DMSO) in capillary tubes and left to slowly settle to the bottom (Supplementary Fig. 1)[11,12]. The LC phases were formed after ~30 days of sedimentation with a volume fraction of ~50%. The sediment exhibited fluidic behavior and could flow upon tilting the capillary tube (Supplementary Fig. 2), in accordance with the colloidal LC behavior reported before[40]. Depending on $R_D$ and $R_L$, the DBCs assembled into a variety of entropy-driven LC mesophases, including BP III, N2, N1, and SmA phases. The long-range ordering and detailed organization of individual DBCs within all the formed phases were characterized by scanning electron microscopy (SEM), in situ confocal microscopy, and polarized optical microscopy (POM), respectively.

For DBCs with two very short e-blocks (e.g., $R_L = 0.10$), the colloids organized into the double-twisted chiral columns, although the constituent DBCs are achiral (Fig. 2a–d). Since the double-twisted chiral columns are randomly oriented and no ordered packing is found, the formed phase of DBCs is BP III phase with a racemic mixture of chiral columns. Within each double-twisted chiral column, the DBCs were rotated in a helical fashion along the long axis ($t_1$) of each column and also formed a barrel-like chiral twist with respect to its layer normal ($t_2$) (Fig. 1). Our in situ confocal microscopy results confirmed that the double-twisted structures were formed in the liquid crystalline state, rather than resulting from the sample drying (Fig. 2c). Though the 2D confocal images (Fig. 2c) are not as clear as the SEM images (Fig. 2a, b), the rotation direction of columns, i.e., the handiness of the columns, can be well defined by eye measurement of 3D confocal images with different depths (as shown in Supplementary Video 1 and 2). Statistically, the populations of right- and left-handed chiral columns are approximately equal by counting 100 columns. Thus, the BP III phase shows locally defined handedness and helical pitch (P) that varies slightly from domain to domain. The half P (1/2P) was determined as

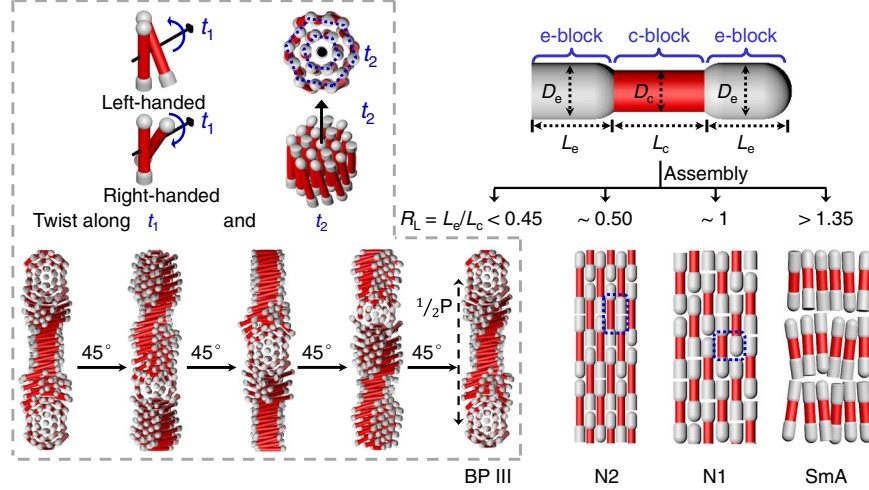

**Fig. 1 | Schematic illustration of four typical LC phases assembled from DBCs.** Schematic illustrations of the geometrical parameters of DBCs and the corresponding assembled BP III, N2, N1, and SmA phases. $D_e$ and $D_c$ are the diameters of the end block and central block, respectively. $L_e$ and $L_c$ are the lengths of the end block and central block, respectively. $R_L$ is the ratio of lengths of end blocks to central blocks. $t_1$ and $t_2$ are the long axis of the chiral column and the layer normal of the barrel-like chiral twist, respectively. 1/2P is the half helical pitch.

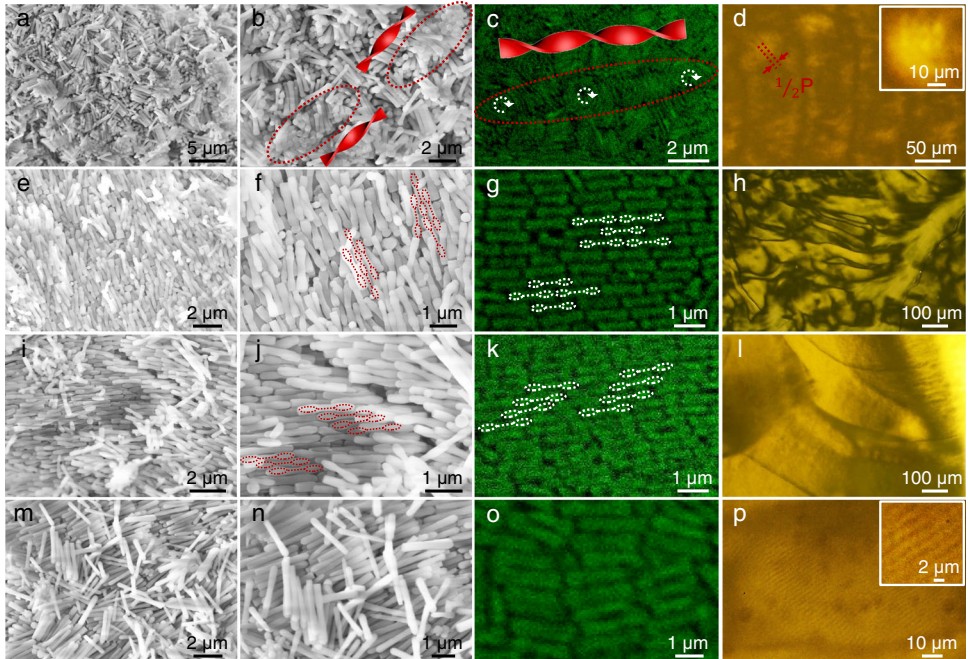

**Fig. 2 | LC phases of DBCs with different geometries.** Representative low (**a**, **e**, **i**, **m**) and high (**b**, **f**, **j**, **n**) magnification SEM images, **c**, **g**, **k**, **o** confocal microscopy images, and **d**, **h**, **l**, **p** POM images of LC phases assembled from different DBCs: **a–d** BP III phase assembled from DBCs with $L_e$ = 160 nm, $L_c$ = 1660 nm, $D_e$ = 315 nm, and $D_c$ = 240 nm, **e–h** N2 phase assembled from DBCs with $L_e$ = 400 nm, $L_c$ = 715 nm, $D_e$ = 280 nm, and $D_c$ = 190 nm, **i–l** N1 phase assembled from DBCs with $L_e$ = 515 nm, $L_c$ = 520 nm, $D_e$ = 235 nm, and $D_c$ = 155 nm, **m–p** SmA phase assembled from DBCs with $L_e$ = 900 nm, $L_c$ = 700 nm, $D_e$ = 245 nm, and $D_c$ = 225 nm.

the distance between two correlative barrel twist with same handedness by counting 100 columns of each BP III sample. For example, the helical structure displays a left-handed column and a right-handed column in Fig. 2b, and a right-handed column in Fig. 2c with 1/2P of 5 ± 0.5 μm (Supplementary Fig. 3). The helical twist can be interrupted when columns with different handedness merge. Within the barrel twist, the central DBC does not tilt with $t_2$, and the tilt angle of DBCs increases from the inner to the outer regime (Fig. 1). The formation of the BP III phase was further confirmed by the characteristic texture in the POM image of periodic lines with a spacing of ~5 μm, corresponding to 1/2P (Fig. 2d)[36,41,42].

The dumbbell shape endows DBCs with interlocking ability. DBCs with two short e-blocks (e.g., $R_L$ = 0.56) assembled into an N2 phase (Fig. 2e–h). In the nematic phase, the degree of orientational order can be described by the 2D order parameter $S_{2d} = \frac{1}{N} \langle \sum_{i=1}^{N} \cos(2\theta_i) \rangle$, where $N$ is the total number of measured DBCs, $\theta$ is the angle between the director and the long axis of each DBC, and the brackets denote an average over all the measured DBCs[43]. As shown in confocal images (Fig. 2g and Supplementary Fig. 4), the DBCs show a high degree of long-range orientational order with $S_{2d}$ ~ 0.74. Moreover, neighboring DBCs are packed in a slipped mode to form two-lock structures; that is, the two thick e-blocks of two end-to-end adjacent DBCs are embedded within the groove (i.e., thin c-block) of a third DBC. The localized POM image (Fig. 2h) shows the typical schlieren texture of the N2 LC phase[44]. DBCs with two intermediate e-blocks (e.g., $R_L$ = 0.99) organized into the N1 phase (Fig. 2i–l). The detailed packing of DBCs is shown in Fig. 2i–k. The concavity of one DBC does not have sufficient room to accommodate two e-blocks of neighboring DBCs to form the two-lock structure. Instead, the groove of each DBC fits one e-block of an adjacent DBC to form the one-lock structure with $S_{2d}$ ~ 0.81 according to the confocal images in Fig. 2k and Supplementary Fig. 5. The N1 phase also exhibits a schlieren texture (Fig. 2l). DBCs with two long e-blocks (e.g., $R_L$ = 1.29) exhibited a SmA phase with the average rod alignment perpendicular to the smectic layer (Fig. 2m–p). The individual DBCs are

located within the smectic layer (as shown in Fig. 2m–o). The formation of the SmA phase was also confirmed by the characteristic striped texture in the POM image (Fig. 2p)[42,44].

## Phase diagram and geometrical effects based on experimental observations

The assembled phases of DBCs were mapped out in a phase diagram in the space of $R_D$ and $R_L$ (Fig. 3a). The achiral DBCs with two very short e-blocks ($R_D$ > 1.1 and $R_L$ < 0.45) underwent spontaneous chiral symmetry breaking, leading to the BP III phase with chiral superstructures (Supplementary Fig. 6). Due to the DBCs' strong tendency to align with each other and the adjacent DBCs not being confined to a planar surface, the steric hindrance between adjacent DBCs induces twists, which eventually generates chiral assembled structures[45]. DBCs can twist in two equivalent ways (Fig. 1), thus leading to the formation of racemic mixtures with equal probability of right- and left-handed chiral domains. It is found that P is strongly dependent on the value of $L_c/D_e$ (Fig. 3b). P increases from 7 to 17 μm with $L_c/D_e$ increasing from 3.0 to 8.5. This work experimentally demonstrated the formation of a double-twisted LC phase from achiral DBCs. This phenomenon is in agreement with previous simulation studies on polymer-tethered nanorods[46], bent-core, and linear rigid particles[4,5].

As $R_L$ increases, DBCs with two short e-blocks (1.3 < $R_D$ < 1.6 and $R_L$ ~0.50) form the N2 phase, as DBCs with two-lock arrangements can pack more efficiently than the simple nematic phase (Supplementary Fig. 7). With the further increase of $R_L$, $L_e$ becomes comparable to $L_c$. The shape complementarity in the morphology leads to the N1 phase with zipper structures (Supplementary Fig. 8)[27] for DBCs with two intermediate e-blocks (1.1 < $R_D$ < 1.6 and $R_L$ ~1.00). Notably, the dashed lines at $R_L$ of 0.50 and 1.00 locate around the centers of N2 and N1 phases, respectively. For DBCs with long e-blocks ($R_L$ > 1.35) or DBCs with small diameter differences ($R_D$ < 1.1), SmA phases are formed due to the mismatch between $L_e$ and $L_c$ of DBCs and the nearly rod-like shape of DBCs (Supplementary Fig. 9)[40]. For DBCs with large diameter difference ($R_D$ > 1.6), isotropic phases (i.e., without orientational and positional order) were found (Supplementary Fig. 10). The formation

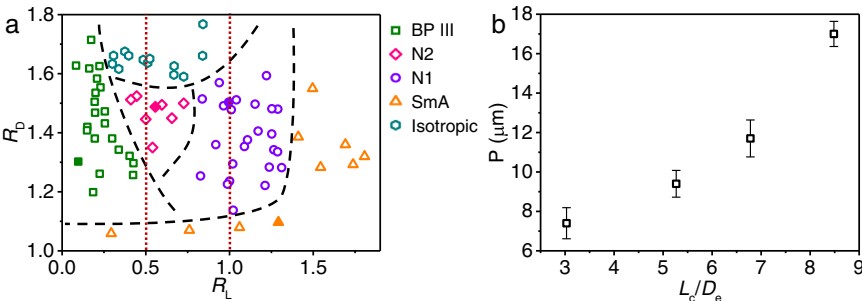

**Fig. 3 | Correlation between geometry and LC phases of DBCs based on experimental results. a** Experimental phase diagram of DBCs as a function of $R_D$ and $R_L$. The solid symbols represent the data points in Fig. 2. The black dashed lines indicate superficial boundaries between phases. The red lines correspond to $R_L =$ 0.5 and $R_L = 1.0$, respectively. They locate approximately at the center of the corresponding phases, namely N2 and N1 phases. **b** Helical pitch P as a function of $L_c/D_e$. P is obtained by counting 100 columns for each sample. Error bar represents standard deviation based on 100 columns.

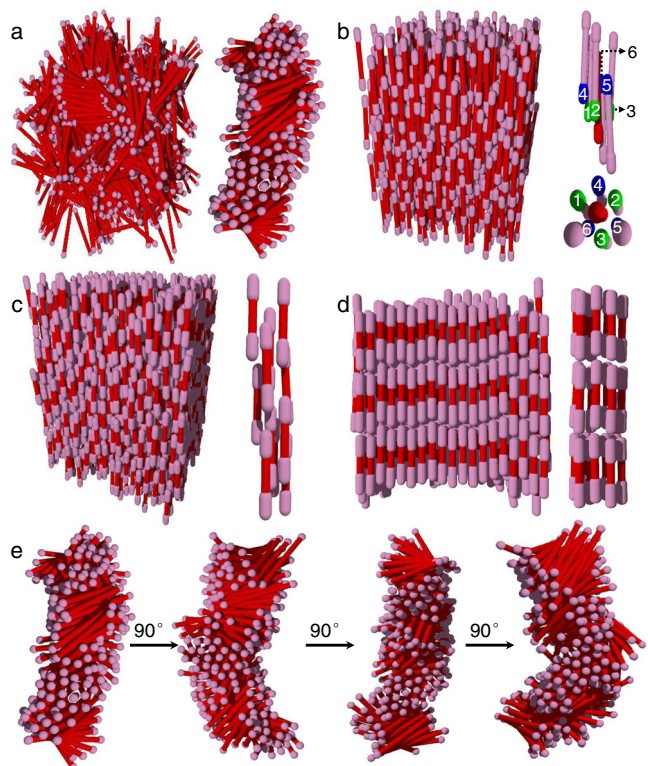

**Fig. 4 | LC phases observed in Brownian dynamics simulations.** Simulation results of LC phases assembled from DBCs with **a** $R_D = 1.40$ and $R_L = 0.15$ in BP III phase (right: a double-twisted column extracted from left), **b** $R_D = 1.40$ and $R_L =$ 0.50 in N2 phase (right: side and bottom view of the typical local structure), **c** $R_D = 1.40$ and $R_L = 1.00$ in N1 phase (right: a cylinder picked out along the $z$ axis), and **d** $R_D = 1.40$ and $R_L = 1.80$ in the SmA phase (right: a cylinder picked out along the $z$ axis). **e** Double-twisted structure viewed from different directions.

of the isotropic phase can be attributed to the large energy barriers for the assembly of DBCs. In summary, these results indicate that subtle variation in the geometry of DBCs has a significant impact on their phase behavior.

## Simulations based on a coarse-grained Brownian dynamics model

We simulated the LC phase formation of DBCs with different geometrical parameters using a coarse-grained Brownian dynamics model (see Methods for simulation details)[47]. Briefly, the DBCs were constructed by fixing linearly arranged rigid spherical beads into a

dumbbell rod geometry in accordance with the size and shape of real DBCs. The e-blocks with a large diameter $D_e$ were located at both ends of the central c-block with a small diameter $D_c$. The interactions between DBCs were set to be purely repulsive Weeks-Chandler-Andersen (WCA) potential[48]. All simulations were carried out in a cubic cell with periodic boundary conditions. To simulate the phase condition with high densities, the length of box size was iteratively increased and decreased upon changing the temperature.

The formation of all the phases observed in the experiments was confirmed by the simulations. Figure 4 shows four characteristic snapshots of the assembled DBCs with different values for $R_D$ and $R_L$. The simulation results indicate that the assembled phases of DBCs are strongly dependent on $R_L$, which is consistent with the experimental results. When $R_D$ was fixed as 1.40, the DBCs with $R_L = 0.15$ accommodated in a BP III phase. Figure 4a, e show the configuration of a single column extracted from the simulation box. The snapshots clearly show that, rather than packing parallel to each other, the DBCs form a regular chiral twist both within and about the column. In this case, the e-blocks prevent regular close packing of the DBCs, which would otherwise lead to the formation of the smectic phase. As a result, the neighboring DBCs are closely packed with a tilted angle to form the twisted structure. It should be noted that these assembled structures exhibit a discontinuity in the twist of the column because of the convergence of columns with different handedness. DBCs with $R_L = 0.50$ form an N2 phase (Fig. 4b). Typically, two e-blocks of two end-to-end adjacent DBCs are trapped in one concave groove of an adjacent DBC. In the local structure of the N2 phase, a total of six DBCs (three green e-blocks above and three blue e-blocks below) packed around the central red c-block. In other words, two layers of e-blocks from surrounding rods are trapped by the central c-block along its long axis. In contrast, DBCs with $R_L = 1.00$ exhibit an N1 phase (Fig. 4c). One e-block of each DBC is preferentially embedded in the concave groove of an adjacent DBC. This configuration allows DBCs to pack closely with neighboring ones, resulting in a zipper structure. As $R_L$ increases, DBCs with $R_L = 1.80$ become stable in the SmA phase (Fig. 4d). In this case, since the e-block is longer than the c-block, the e-block cannot be fit into the concave groove of a DBC. Therefore, the effect of the diameter difference of blocks becomes negligible, making them behave like conventional colloidal rods[40]. Finally, it was found that only DBCs with $1.1 \leq R_D < 1.6$ could exhibit assembly behavior that are distinct from rod-like colloids. DBCs with even larger $R_D$ ($\geq 1.6$) tend to form an isotropic phase (i.e., without positional and orientational order) due to the high energy required for the assembly of DBCs (Supplementary Fig. 11).

## Phase diagram and geometrical effects based on simulations

We evaluated the formation and transitions of mesophases by systematically varying $R_D$ and $R_L$ in simulations. The results are

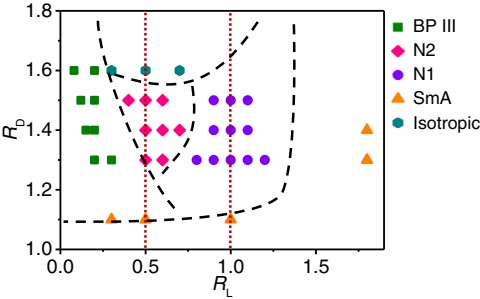

**Fig. 5 | Phase diagram based on simulation results.** Computational phase diagram of DBCs as a function of $R_D$ and $R_L$. The boundary dashed lines are taken from the experimental phase diagram.

summarized in the phase diagram shown in Fig. 5. The phase regions are in good agreement with the experimental phase diagram. The formation of mesophases is determined by both $R_D$ and $R_L$. The BP III phase is stable only when $R_L$ is small, i.e., the DBCs possess two very short e-blocks (Supplementary Fig. 12). Chirality of the structure arises from the twist arrangement of achiral DBCs. When the e-block becomes longer the formation of the BP III phase is suppressed, while the formation of N2 and N1 phases is strongly promoted. The N2 phase is formed when $R_L$ is around 0.5 (i.e., the length of c-block is approximately twice of that of e-block). In this case, the concave groove of a DBC can accommodate two e-blocks in the long axis direction, which is driven by excluded volume interactions (Supplementary Fig. 13). The N1 phase was observed with further increasing $R_L$ to about 1.0. The comparable length of c-block to e-block enables the trapping of one e-block in each concave groove of DBCs in the long axis direction. When the e-block is longer than the c-block, the DBCs form the SmA phase.

### LC Phase manipulation with magnetic fields

Thanks to the positive diamagnetic anisotropy of DBCs, the colloidal LCs of DBCs are responsive to external magnetic fields, and the LC phases can be dynamically tuned by controlling the DBC orientation. Upon applying a magnetic field of 5 T, DBCs with two very short e-blocks ($R_D > 1.1$ and $R_L < 0.45$) were rotated and aligned their long axis along the field direction, resulting in their visual texture changing from BP III to regular nematic phase as shown in the POM images (Supplementary Fig. 14). The strip pattern of nematic phase is well-aligned to the magnetic field. In addition, fewer defects were observed after the application of the magnetic field, indicating the efficient alignment of DBCs by the magnetic field. This result suggests the potential of such DBC-based colloidal phases for application in stimuli-responsive and magnetically switchable systems.

## Discussion

In conclusion, we have systematically studied the phase behavior of the colloidal analogs of dumbbell-shaped molecules, so-called DBCs. Both experimental and simulation results confirmed that $R_D$ and $R_L$ are critical for the formation of mesoscopically ordered phases. Our experimental results showed that DBCs with $R_D$ of 1.0–1.8 and $R_L$ of 0.1–1.8 can be assembled into a variety of LC phases including BP III, N2, N1, and SmA phases. The phases and onset of phase transitions were found to be strongly dependent on $R_D$ and $R_L$ of the DBCs. It is worth noting that double-twisted chiral superstructures with racemic mixtures, namely BP III, can be produced from rigid achiral colloids in the range of $R_D > 1.1$ and $R_L < 0.45$. Brownian dynamics simulation was used to further establish the correlation between the geometry and phase formation of DBCs. The simulation results are in good agreement with the experimental data, providing a theoretical basis for the future

design of complex mesogenic systems. In addition, we show that the colloidal LC phases can be dynamically tuned by controlling the orientation of DBCs with magnetic fields. This work provides insights into the rules required for the design of particles of different shapes to achieve superstructures with increased complexity and desired stimuli-responsive electronic and photonic properties.

## Methods

### Synthesis of DBCs

The silica DBCs were synthesized by a recently developed emulsion-templated wet-chemical approach[30]. In a typical experiment, 500 mg polyvinylpyrrolidone ($Mw = 40,000$, Sigma-Aldrich) was dissolved in 5.0 ml 1-pentanol (99%, Sigma-Aldrich) in a 10 ml glass vial under sonication. 140 µl deionized water (18.2 MΩ), 50 µl 0.18 M sodium citrate dihydrate (99%, Sigma-Aldrich) aqueous solution, 475 µl anhydrous ethanol (Pharmco-Aaper), and 100 µl ammonium hydroxide solution (28 wt%, Sigma-Aldrich) were added to the glass vial and vortexed for 1 min. After the mixture was left standing for 5 min to release gas bubbles, 50 µl tetraethylorthosilicate (TEOS, 98%, Sigma-Aldrich) was added and the solution was then gently shaken for 30 s. The $L_e$ (160−900 nm) and $L_c$ (170−3360 nm) of DBCs can be individually controlled by tuning the reaction time during the growth of each block. The $D_e$ of DBCs can be tuned from 190 to 460 nm by adjusting the amount of ammonium hydroxide. The $D_c$ of DBCs can be adjusted in the range of 140−340 nm by varying the strength of perturbation in the reaction temperature (25−70 °C) during the DBC growth. A higher temperature during the perturbation period produces a smaller $D_c$. For example, for DBCs with $L_e = 160$ nm, $L_c = 1660$ nm, $D_e = 315$ nm, and $D_c = 240$ nm, the reaction mixture was incubated at 25 °C for 30 min, 60 °C for 20 min, and 25 °C for another 4 h.

The fluorescent DBCs were prepared by the subsequent growth of a fluorescent inner shell and a non-fluorescent outer shell. To grow a fluorescent shell, the as-prepared DBCs were dispersed into 5.0 ml of ethanol with 330 µl of deionized water and 400 µl of ammonium hydroxide solution (28 wt%). Then a solution containing 167 µl of ethanol, 20 µl of TEOS, 0.8 mg of fluorescein isothiocyanate (FITC, isomer I, 98%, Sigma-Aldrich), and 1.0 µl of 3-aminopropyltriethoxysilane (APS, 99%, Sigma-Aldrich) was added. The reaction was sonicated at 25 °C for 4 h. Afterwards, the DBCs were washed three times with ethonal. The non-fluorecent shell was grown via a similar procedure but without the addition of APS and FITC. The thicknesses of the fluorescent shell and non-fluorescent shell are both *ca.* 60-80 nm.

### Purification

The as-synthesized DBCs were purified by centrifugation at 8500×*g* for 10 min. The precipitated DBCs were then washed three times with ethanol under centrifugation at 4000×*g*. Finally, the DBCs were suspended in ethanol and centrifuged at 1500×*g* for 10 min to remove small particles and other lightweight impurities.

### Assembly of DBCs into LCs

The DBC dispersions in DMSO (99.9%, Fisher Chemical) with a volume fraction of 20% were filled into one-end-sealed glass capillaries (Stuart) with an outside diameter of 1.9 mm, an inside diameter of 1.3 mm, and a length of 100 mm, and rectangular capillaries (VitroCom) with 2.0 mm in width, 0.2 mm in inner-thickness, 0.14 mm in wall thickness, and 50 mm in length. The other ends of the capillaries were then sealed by melting. Subsequently, the capillaries were left in a vertical position for the sedimentation of DBCs. After one month, the capillaries were opened, and the sediment was slowly dried in air at room temperature and taken out for SEM characterizations. The samples in rectangular capillaries were directly used for POM and confocal microscopy characterization.

 

## Orientational order parameter

Because the colloidal LCs were formed under gravity and most colloids oriented along gravity direction, we, therefore, defined the gravity direction as the global orientation. We have manually measured the angles between the director and the long axis of each DBC for 200 DBCs of each sample using Image J software that is available at NIH Web site, to calculate $S_{2d}$ for N1 and N2 phase, respectively.

## Phase switching with magnetic fields

The switching of colloidal LC phases was conducted under a magnetic field generated by 9 T Room Temperature Bore Magnet System (Cryomagnetics.Inc). The samples in rectangular capillaries upon sedimentation were allowed to sit undisturbed for 1 h under a magnetic field of 5 T and then examined using POM.

## Characterizations

The SEM images of DBCs and their assemblies were taken using a Hitachi SU-70 Schottky field emission gun SEM at an operation voltage of 10 kV. The confocal microscopy images with a 300 nm voxel depth were acquired using an Olympus FV3000 inverted microscope equipped with a ×60 (NA = 1.42) oil immersion objective (Olympus) and a 532 nm laser. POM images were obtained by an Olympus BX50F4 Microscope with crossed polarizing filters.

## Computer simulations

Computer simulations on the formation mechanism of different structures of DBCs were approached as follow: the e-block and c-block with the diameters of $D_e$ and $D_c$ were constructed by fixing linearly arranged spherical beads, respectively. Two e-blocks with a diameter of $D_e$ were located at both ends of a DBC. Each DBC was treated as a rigid body in the simulation. Non-bonding interactions between DBCs were described by the WCA potential. The WCA potential is the 12-6 Lennard-Jones potential truncated at the energy minimum and shifted vertically by $\varepsilon/k_B T$; it is a purely repulsive potential featuring the volume occupied by DBCs. The Brownian dynamics of rigid bodies in the canonical ensemble were adopted for the simulations. The details of the simulation method can be found in ref. 38. The length and the energy units of the simulation system are $\sigma$ and $\varepsilon$, respectively. The time step is in a unit of $\sigma\sqrt{m/T}$, while the unit of temperature is $\varepsilon/k_B$. A leapfrog algorithm of rigid bodies with a time step of $\Delta t = 0.01$ was adopted to integrate the equations of motion[49]. All the simulations were performed using the GALAMOST package[50,51].

In the simulations, DBCs in the initial configuration were regularly generated in the simulation box with a volume fraction of 0.48. The regular initial configuration is helpful for testing whether the smectic structure is stable under certain simulation conditions and getting the physical insight of the formation mechanism of different structures. Then the temperature of the system was iteratively increased and decreased between $0.5\,k_B/\varepsilon$ and $1.0\,k_B/\varepsilon$ with an interval of $\Delta T = 0.05\,k_B/\varepsilon$ every $2 \times 10^6$ timesteps. The box size in z-direction was also iteratively increased and decreased with the variance of temperature. The volume fraction was varied between 0.46 and 0.48. Finally, the system was further equilibrated for $1 \times 10^7$ timesteps at the 0.48 volume fraction[52]. In the present work, the parameter $L/D_e$ was fixed at around five except for some special cases. This value corresponds to a moderate rod length according to our previous work[38]. The values of $L/D_e$ in several systems with extremely small $R_L$ (i.e., 0.08, 0.12, and 0.15) were set as 11.5, 7.0, and 8.1, respectively.

## Data availability

The authors declare that the data supporting the findings of this study are available within the article and its Supplementary Information files.

## Code availability

Code used to generate the results in the paper can be downloaded from Github (github.com/PeiHanwen/Brownian-Dynamics/tree/bdrelease).

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

## Acknowledgements

We acknowledge the support of the State Key Laboratory of Supramolecular Structure and Materials, and the State Key Laboratory of Molecular Engineering of Polymers. We thank the National Natural Science Foundation of China (22122203, 51973038, 21534004, 91963107, 21911530179, and 52073114), the National Key R&D Program of China (2018YFB0703701), the Program for JLU Science and Technology Innovative Research Team (JLUSTIRT2017TD-06), the Key-Area Research and Development Program of Guangdong Province (X200281AL200), and the Innovation Program of Shanghai Municipal Education Commission (2021-01-07-00-07-E00073) for financial support.

## Author contributions

G.C., Y.Y., Z.L., K.L., and Z.N. designed the research. G.C. and Y.Y. performed experiments. H.P. performed simulations. Y.Y., G.C., H.P., X.Z., W.S., Y.Z., Z.L., K.L., and Z.N. analyzed the data. M.L., C.F., and B.Y. provided valuable suggestions for the discussion of the results. G.C., Y.Y., Z.L., K.L., and Z.N. prepared the manuscript. All the authors read and commented on the manuscript.

## Competing interests

The Authors declare no competing interests.
