## [Peer Review File · Nature Communications]

REVIEWER COMMENTS

Reviewer #1 (Remarks to the Author):

This paper reports the study of the liquid crystal phase behavior of dumbbell-shaped colloids. The authors use both experiments and Brownian dynamics simulations to systematically study the impact of particle geometry on the self-assembly of colloidal dumbbells into LC phases. In particular, they use SEM and POM to (experimentally) characterise the structure of the LC phases exhibited by the particles. They report the finding of four main LC phases, and claim that one of them shows chiral ordering. Finally, they apply an external magnetic field to control the alignment the colloidal particles along a given direction. The main point of the paper seems to be that chiral structures resembling to blue phases can be formed by achiral dumbbell-shaped colloidal building blocks.

Although the particles used in this work present some interesting liquid crystal behavior and its dependance on the geometry of the colloids, I am not convinced that the results shown are sound enough to be published in Nature Communications. My major concern is that there is not enough quantitative experimental data to support the claims the authors make about the observation of four distinct LC phases, in special the chiral LC phase. In my opinion, the following points are not clear:

1) In general, all the SEM images presented in Fig 2 -- used to characterise the structure of the different LC phases at the single particle level--, are notoriously difficult to interpret. While some sort of orientational order can be observed in the N1, N2 and SmA phases, the helical structures for N* are just very difficult (if not impossible) to observe from the SEM images. In other words, by just looking at these images (and not comparing it to any of the dotted line overlays or to the schematics shown in Fig 1.), I was unable to observe or identify any particular LC order. In fact, some of the dotted line drawings (Fig2.e) do not even correspond to physical particles, which I find quite misleading. Moreover, the fact that there is a rather large number of particles which appear randomly oriented in all of the images, which presumably come from the SEM sample preparation (and if it is the case, it should be noted), make even more difficult the visualisation of the structure of these phases.

2) The authors use POM to help identifying/classifying the four different claimed LCs phases. While birefringence is observed in all four samples, confirming the LC nature of these phases, the authors constantly refer to a 'characteristic' or 'typical' textures for each of the phases, but no citations are provided to compare such textures to textures documented in literature. Moreover, the POM texture of N* does not really show periodic lines every 5 microns, if anything, vertical lines seems to appear spaced out about 60 microns. Also, what does the inset in the same panel (Fig2.c) show?. In the case of the SmA, the stripped texture shown in Fig2.I seems to be more related to the real space structure of the phase rather than the interaction of polarized light with a SmA phase (see book "Liquid crystals" by Peter J. Collings). Therefore, I think that in this system the POM data provides extra characterisation of

the colloidal LC phases, but it is far from being a finger-print demonstration of a certain type of order if proper citations are not provided.

3) In particular, I am worried that the characterisation of the double-twist chiral phase is not very rigorous. How is $1/2P$ measured? From the SEM images? In that case, the plane of the phase modulation should coincide with the plane of observation, which seems quite unlikely -- although marginally possible to happen--. Some sort of three-dimensional characterisation would be needed to confirm the such chiral ordering in space (perhaps use 3D confocal microscopy?). Also, regarding Fig.3d, how many times is P determined? i.e. where are the error bars coming from? As mentioned in point (2), I find that the determination of P from SEM measurements would not be ideal. Moreover, the units for P in this plot should be in micrometers (μm) and not nanometers (nm).

4) Also, the experimental procedure used to measure the orientational order parameter S_2 for N_1 and N_2 is poorly explained: Are the SEM images used to calculate such parameter? In that case how do they calculate the global orientation? And the S_2 value for every single DBC? (in the SEM images particles are on the top of each other). A different approach to measure S_2 would be required (using for example 2D confocal snapshot).

Finally, I find the abstract and introduction of the paper is not very well-structured and a bit misleading. It is not clear why is of 'paramount importance' or 'urgent' the study of the formation of chiral structures using achiral colloids. It would be useful to explain that, for example, colloidal liquid crystals formed by dumbbells would enable the visualisation the structure of these phases at the single particle level, which could help unraveling how chirality can emerge from achiral building blocks. For all of these reasons I do not recommend publication of this manuscript in Nature Communications, but I encourage the authors to continue working on it to provide more evidence to complete this piece of work.

Further comments and questions:

- In general, the introduction lacks of citations: e.g. end of first or second line.
- The concept of blue phase is not explained.
- Line 57-58: "chiral LCs can be formed from the chiral organization of achiral objects (e.g., molecules and colloids) that do not have a chiral center". Re-phrase to: 'chiral LCs can be formed from the chiral organization of objects (e.g., molecules and colloids) that do not have a chiral center'.
- Line 62: the nematic twist bend phase formed by banana-shaped molecules has been extensively reported as well (see e.g., Pajak, PNAS 2018 or Chen, PRE (2014)).
- Line 69: molecular liquid crystals can definitely studied quantitatively, studies of their structure at single molecule level are the ones prohibitively difficult.

- Line 72: “concave geometry” is quite general, perhaps you could specify a little bit more the geometry of the particles.
- The notation N^* is typically used in literature to refer to a chiral nematic phase.
- Line 96: Give the units for RD and RL in between brackets.
- It would be useful to note somewhere at the beginning of the paper that the particles used in this work are made out of silica.
- Figure 4.e is not mentioned anywhere in the main text.
- Perhaps it would help the reader to compare the experimental and simulated phase diagram in the same figure.
- While the section where the magnetic field manipulation shows the successful external manipulation of LCs formed by colloidal dumbbells, it feels it’s breaking the flow of the paper as it does not add a lot to its main message, which would be the phase behavior of colloidal dumbbells and formation of chiral phases.

Reviewer #2 (Remarks to the Author):

This manuscript describes observations of the phase behaviour of colloids in the shape of dumbbells, i.e. rod-shaped microparticles with broadened segments at either end. Colloidal rods are known to spontaneously form liquid crystal (LC) phases that are analogous to the LCs formed by their molecular counterparts, but can be studied on the level of individual particles. Moreover, their shapes and interactions can be varied, making it possible to determine the effects of such parameters on the LC phases formed. New approaches to precisely shape colloidal rods have now made it possible to prepare dumbbells. This is an important model system that has been considered in many theoretical and simulation studies, especially because of the possibility of the formation of a chiral phase from these achiral particles. The authors use these particles to study the phase behavior as a function of shape (the diameters and lengths of the segments). They convincingly confirm the existence of such a phase in experiment, besides several other phases, and map out the phase diagram. These results are impressive and of high importance. They are on the whole well supported by the experimental evidence, however, I will raise a few comments below. I expect this work will serve as a benchmark for future studies, both in the molecular and colloidal field. The work is also of considerable interest for the development of switchable optical materials. I recommend acceptance of this manuscript, but think that a number of points should be elaborated first:

1) The evidence (shown in Figure 2, as well as in SI) for the N-star and N1 phases is compelling. However, I found it difficult to recognize the N2 structure. There seem to be only a few isolated groups of particles arranged in the proposed structure. Could the authors provide more convincing images?

2) The particles in Figures 2j and 2k don't have a clear dumbbell shape, as R_D is very close to unity. It would be much more convincing if the authors showed an SEM image of a phase formed by rods with a large R_d (corresponding to one of the other points in Fig. 3).

3) I am surprised that the authors didn't mention that the colloidal dumbbells are made of silica. Not in the introduction, nor in the results, and not even in the methods does the word silica appear. This is a curious omission.

4) As evidence of the formation of colloidal LC phases it is mentioned (on page 4) that these phase behaved as a fluid when tilted. But the authors (and readers) should be aware that even colloidal crystals (that is, colloidal solids) display such fluid behaviour as a result of their extremely low yield stress.

5) The nematic N1 and N2 phases seem to be rather uncommon. Have they been observed before? Since nematic phases are supposed to have only orientational order and no positional order can these really be called nematics? Please discuss.

6) Please discuss how the orientational order parameter is determined from SEM images.

7) The authors refer to the POM images with phrases such as "the characteristic texture" or "typical schlieren texture" without further explanation. It would be helpful to interpret these images with further explanations, or provide references.

8) In Figure 4d, aren't the smectic layers tilted with respect to the director? In that case it should not be called Smectic A, but rather Smectic C.

9) In the experiment using a magnetic field to orient the LC phase, was the change reversible? This would be important to the applications that the authors mention.

10) In the Methods section I assume the particles were first transferred from ethanol to DMSO by centrifugation? I also assume that this was pure DMSO and the colloids are charge stabilized. What can the authors say about the softness of particle interactions? What was the length of the electric double layer in the solvent? Was the ionic strength measured? This is important for a better comparison with theory/simulations.

11) In the Methods it is stated that “the sediment was taken out and slowly dried in air”. Wasn’t this the other way around?

12) The word “freezing” in the phrase “the DBCs were constructed by freezing linearly arranged...” is unclear. Please rephrase.

Reviewer #3 (Remarks to the Author):

The paper by Yang Yang et al. deals with the formation of liquid crystalline phase from dumbbell-shaped colloids. Specifically, one of their claims is the evidence of the chiral organization reminiscent of double twist cylinders observed in blue phases from their achiral particles. Note that for steric reason, it is expected to get twist between the particles so the result is not for me highly surprising. Interestingly, the authors obtain also other mesophases by tuning the geometrical features of their particles, including two nematic phases (one-lock (N1) and two-lock (N2) structures) and one smectic phase. However, the experimental proofs of such claims is rather poor for a publication in a journal like Nature Communications: beyond low magnification polarizing microscopy (which only gives indirect structural information and nothing at the single particle level), only scanning electron microscopy has been performed to give insights about the liquid crystalline organization at the particle scale. However such an electron microscopy technique is intrinsically obtained on dried samples. Direct observations of particles in suspensions, which means at equilibrium, seem feasible, as shown both by the micrometer size of their particles and by their possible labeling by FITC dyes (as indicated in Supplementary Figure 2). Therefore I strongly recommend to the authors to partially label some of dumbbell-shaped colloids and to perform extensive study of their suspensions by optical confocal microscopy in order to get direct structural demonstration of their liquid crystalline phase at equilibrium, as well as some information of the dynamics of the system.

Beyond this key-comment, please find below other points that have to be addressed:

1) The absence of overall chirality in the suspensions should be proved by providing proper statistics (See line 109 p.5), and not only a claim based on one (or a few) image(s).

2) Still dealing with chirality, I guess that the unit of the helical pitch in Figure 3b is in μm (and not in nm as it is written).

3) Why the authors have not varied the initial volume fraction of their colloids to prepare their samples? Mostly the geometric features of their dumbbells have been changed (See Fig. 3a) but it would be as important to get the dependence of the phase behavior as a function of the particle volume fraction (for a given size).

4) Some claims in the abstract and in the introduction are clearly overstated. For instance, it is mentioned the blue phase based LCDs which are indeed interesting for their fast response. However, mentioning LCD based on colloidal particles is certainly awkward, considering the very slow response time of colloids, the difficulty of applying electric fields in colloidal suspensions, etc...

REVIEWER COMMENTS

Reviewer #1 (Remarks to the Author):

General comments: Although the particles used in this work present some interesting liquid crystal behavior and its dependence on the geometry of the colloids, I am not convinced that the results shown are sound enough to be published in *Nature Communications*. My major concern is that there is not enough quantitative experimental data to support the claims the authors make about the observation of four distinct LC phases, in special the chiral LC phase.

Re: We thank the reviewer for the insightful comments. We agree with the reviewer that in our original submission the experimental data were not quantitative enough to clearly present the LC phases. In response, we have performed additional experiments and made substantial changes to clearly present the LC phases in the revised manuscript. Please see our detailed response to the comments as follows.

Comment 1: In general, all the SEM images presented in Fig 2 -- used to characterise the structure of the different LC phases at the single particle level--, are notoriously difficult to interpret. While some sort of orientational order can be observed in the N1, N2 and SmA phases, the helical structures for N* are just very difficult (if not impossible) to observe from the SEM images. In other words, by just looking at these images (and not comparing it to any of the dotted line overlays or to the schematics shown in Fig 1.), I was unable to observe or identify any particular LC order. In fact, some of the dotted line drawings (Fig2.e) do not even correspond to physical particles, which I find quite misleading. Moreover, the fact that there is a rather large number of particles which appear randomly oriented in all of the images, which presumably come from the SEM sample preparation (and if it is the case, it should be noted), make even more difficult the visualisation of the structure of these phases.

Re: We are sorry that the structural characterizations in our original submission were not as clear as they should be. In response, we have used *in situ* 3D confocal microscopy to further confirm the structures of four LC phases, in addition to SEM and POM characterizations. The confocal microscopy results presented the *in situ* assembled structures of different phases in the liquid crystalline state, which are consistent with our SEM and POM results of the corresponding dried samples. The new results have been included as a new Fig. 2 in the revised manuscript, new Supplementary Fig. 3-5 in the revised Supplementary Information, and new Supplementary Video 1 and 2. We have also added the following statements accordingly in the revised manuscript (pp4): ‘... Our *in situ* 3D confocal microscopy results confirmed that the double-twisted structures were formed in the liquid crystalline state, rather than resulting from the sample drying (Fig. 2c). The handedness of the columns can be identified from the 3D confocal images of different depths (Supplementary Video 1 and 2).’.

Figure 2 | LC phases of DBCs with different geometries. Representative low (**a,e,i,m**) and high (**b,f,j,n**) magnification SEM images, (**c,g,k,o**) confocal microscopy images, and (**d,h,l,p**) POM images of LC phases assembled from different DBCs: (**a,b,c,d**) BP phase assembled from DBCs with $L_e = 160$ nm, $L_c = 1660$ nm, $D_e = 315$ nm, and $D_c = 240$ nm, (**e,f,g,h**) N2 phase assembled from DBCs with $L_e = 400$ nm, $L_c = 715$ nm, $D_e = 280$ nm, and $D_c = 190$ nm, (**i,j,k,l**) N1 phase assembled from DBCs with $L_e = 515$ nm, $L_c = 520$ nm, $D_e = 235$ nm, and $D_c = 155$ nm, (**m,n,o,p**) SmA phase assembled from DBCs with $L_e = 900$ nm, $L_c = 700$ nm, $D_e = 245$ nm, and $D_c = 225$ nm.

Comment 2: The authors use POM to help identifying/classifying the four different claimed LCs phases. While birefringence is observed in all four samples, confirming the LC nature of these phases, the authors constantly refer to a ‘characteristic’ or ‘typical’ textures for each of the phases, but no citations are provided to compare such textures to textures documented in literature. Moreover, the POM texture of N* does not really show periodic lines every 5 microns, if anything, vertical lines seems to appear spaced out about 60 microns. Also, what does the inset in the same panel (Fig2.c) show?. In the case of the SmA, the stripped texture shown in Fig2.I seems to be more related to the real space structure of the phase rather than the interaction of polarized light with a SmA phase (see book ‘Liquid crystals’ by Peter J. Collings). Therefore, I think that in this system the POM data provides extra characterisation of the colloidal LC phases, but it is far from being a finger-print demonstration of a certain type of order if proper citations are not provided.

Re: We thank the reviewer for the insightful comment. In response, we have added citations to the description of each texture (References 32, 36, and new 37 and 39). Due to the much larger dimension of colloidal particles (a few microns) than molecules (a few nanometers), the textures of the colloidal LCs are usually different from that of the molecular LCs with the same phase structure. According to the previous reports on colloidal LCs, the chiral nematic phase usually shows periodic line texture (References 32, 36, and 37), the nematic phase often exhibits schlieren texture (N1 and

N2 phase, Reference 39), and the SmA phase frequently displays stripped texture (References 37 and 39). The inset of Fig. 2c shows the periodic lines of 5 microns in spacing. The vertical lines in Fig. 2c are often seen in BP phase, but, unfortunately, we do not know what they exactly represent at this stage. We also agree with the reviewer that the stripped texture in Fig. 2l is related to the real space structure, due to the large size of colloidal particles.

***Comment 3:** In particular, I am worried that the characterisation of the double-twist chiral phase is not very rigorous. How is $1/2P$ measured? From the SEM images? In that case, the plane of the phase modulation should coincide with the plane of observation, which seems quite unlikely -- although marginally possible to happen--. Some sort of three-dimensional characterisation would be needed to confirm the such chiral ordering in space (perhaps use 3D confocal microscopy?). Also, regarding Fig.3d, how many times is P determined? i.e. where are the error bars coming from? As mentioned in point (2), I find that the determination of P from SEM measurements would not be ideal. Moreover, the units for P in this plot should be in micrometers (μm) and not nanometers (nm).*

Re: We thank the reviewer for the helpful comment and suggestion. As the reviewer suggested, we have performed 3D confocal microscopy to characterize the BP phase as shown in the new Fig. 2, Supplementary Fig. 3, Supplementary Video 1 and 2. The half P ($1/2P$) was determined as the distance between two correlative barrel twist with same handedness by counting 100 columns of each BP sample in the 3D confocal microscopy images. For example, $1/2P$ was measured to be $5 \pm 0.5 \mu\text{m}$ in Fig. 2c and Supplementary Fig. 3. The $1/2P$ value determined from confocal microscopy images and SEM images is in good agreement. Accordingly, we have added the following statements in the revised manuscript (pp5): ‘...The half P ($1/2P$) was determined as the distance between two correlative barrel twist with same handedness by counting 100 columns of each BP sample.’. (pp5): ‘..., and a right-handed column in Fig. 2c with $1/2P$ of $5 \pm 0.5 \mu\text{m}$ (Supplementary Fig. 3).’.

The unit in Figure 3b was in micrometers. We have corrected the units in Figure 3b.

Supplementary Figure 3. Quantification of the $1/2P$ value by analyzing confocal microscopy images. (a) Ten representative confocal microscopy images of two correlative barrel twist with same handedness assembled from DBCs with $L_e = 160$ nm, $L_c = 1660$ nm, $D_e = 315$ nm, and $D_c = 240$ nm. (b) $1/2P$ distribution is obtained by counting 100 columns.

Comment 4: Also, the experimental procedure used to measure the orientational order parameter S_2 for N_1 and N_2 is poorly explained: Are the SEM images used to calculate such parameter? In

that case how do they calculate the global orientation? And the S_2 value for every single DBC? (in the SEM images particles are on the top of each other). A different approach to measure S_2 would be required (using for example 2D confocal snapshot).

Re: We thank the reviewer for the constructive comment. In response, we have performed 2D confocal microscopy to measure S_2 and the new results are included in the new Fig. 2, Supplementary Fig. 4 and Supplementary Fig. 5. Because the colloidal liquid crystals were formed under gravity and most colloids oriented along gravity direction, we, therefore, defined the gravity direction as the global orientation. We have measured 200 DBCs at different areas. Accordingly, we have added the following statements in the revised manuscript (pp5): ‘...In the nematic phase, the degree of orientational order can be described by the 2D order parameter $S_{2d} = \frac{1}{N} \langle \sum_{i=1}^N \cos(2\theta_i) \rangle$, where N is the total number of measured DBCs, θ is the angle between the director and the long axis of each DBC, and the brackets denote an average over all the measured DBCs³⁸. As shown in confocal images (Fig. 2g and Supplementary Fig. 4), the DBCs show a high degree of long-range orientational order with $S_{2d} \sim 0.74$.’ (pp5): ‘...with $S_{2d} \sim 0.81$ according to the confocal images in Fig. 2k and Supplementary Fig. 5.’ (pp10): ‘...Orientational order parameter. Because the colloidal liquid crystals were formed under gravity and most colloids oriented along gravity direction, we, therefore, defined the gravity direction as the global orientation. We have measured 200 DBCs of each sample at different areas to calculate S_{2d} for N1 and N2 phase, respectively.’

Supplementary Figure 4. The measurement of S_{2d} . (a) 10 representative confocal microscopy images of N2 phase assembled from DBCs with $L_e = 400$ nm, $L_c = 715$ nm, $D_e = 280$ nm, and $D_c = 190$ nm. (b) $\cos(2\theta)$ distribution is obtained by measuring 200 DBCs. The arrow represents the direction of gravity.

Supplementary Figure 5. The measurement of S_{2d} . (a) 10 representative confocal microscopy images of N1 phase assembled from DBCs with $L_e = 515$ nm, $L_c = 520$ nm, $D_e = 235$ nm, and $D_c = 155$ nm. (b) $\cos(2\theta)$ distribution is obtained by measuring 200 DBCs. The arrow represents the direction of gravity.

Comment 5: Finally, I find the abstract and introduction of the paper is not very well-structured and a bit misleading. It is not clear why is of ‘paramount importance’ or ‘urgent’ the study of the formation of chiral structures using achiral colloids. It would be useful to explain that, for example, colloidal liquid crystals formed by dumbbells would enable the visualisation the structure of these phases at the single particle level, which could help unraveling how chirality can emerge from achiral building blocks. For all of these reasons I do not recommend publication of this manuscript in Nature Communications, but I encourage the authors to continue working on it to provide more evidence to complete this piece of work.

Re: We thank the reviewer for the insightful comment. In response, we have revised the abstract and introduction. Accordingly, we have added the following statement in the revised manuscript (pp2): ‘...Chiral LC blue phases (BPs) are attractive for use in fast optical displays and electrooptical devices, but the construction of BPs is limited to a few chiral building blocks and the formation of BPs from achiral ones is often counterintuitive’. (pp3): ‘...Blue phase (BP) LC is one of the most fascinating chiral LCs for applications in fast light modulators or tunable photonic crystals^{2,3}. BPs with a network of double-twisted columns exist in a small temperature range between the isotropic and chiral nematic phases. There are three types of BPs, namely, BP I, BP II, and BP III. BP I and BP II possess body-centered and simple cubic structures, respectively, whereas BP III have amorphous structure. Despite reasonable progress in the field, several barriers remain in the applications of BPs: (i) BPs only exist in a very narrow temperature range (usually less than a few kelvin)^{4,5}, (ii) only a few chiral building blocks can form BPs, and (iii) the molecular origin for the chirality of BPs is still unclear^{2,3}’. (pp3): ‘..., to allow one to deduce a link between microscopic and macroscopic properties.’. (pp3): ‘...Colloidal LCs formed by DBCs would enable the visualization of the twisted structure at the single particle level, thus providing new insights into how chirality can emerge from achiral building blocks.’.

Further comments and questions:

- In general, the introduction lacks of citations: e.g. end of first or second line.

Re: The new citations have been added in the introduction (References 1–5).

- The concept of blue phase is not explained.

Re: The concept of blue phase has been explained. Accordingly, we have added the following statement in the revised manuscript (pp3): ‘...Blue phase (BP) LC is one of the most fascinating chiral LCs for applications in fast light modulators or tunable photonic crystals^{2,3}. BPs with a network of double-twisted columns exist in a small temperature range between the isotropic and chiral nematic phases. There are three types of BPs, namely, BP I, BP II, and BP III. BP I and BP II possess body-centered and simple cubic structures, respectively, whereas BP III have amorphous structure. Despite reasonable progress in the field, several barriers remain in the applications of BPs: (i) BPs only exist in a very narrow temperature range (usually less than a few kelvin)^{4,5}, (ii) only a few chiral building blocks can form BPs, and (iii) the molecular origin for the chirality of BPs is still unclear^{2,3}’.

- Line 57-58: ‘chiral LCs can be formed from the chiral organization of achiral objects (e.g., molecules and colloids) that do not have a chiral center’. Re-phrase to: ‘chiral LCs can be formed from the chiral organization of objects (e.g., molecules and colloids) that do not have a chiral center’.

Re: The introduction has been revised and the above sentence has been removed in the revised manuscript. Please see more details in our response to *Comment 5* by the *Reviewer 1*.

• Line 62: the nematic twist bend phase formed by banana-shaped molecules has been extensively reported as well (see e.g., Pajak, PNAS 2018 or Chen, PRE (2014)).

Re: The introduction has been revised in the revised manuscript. Please see more details in our response to *Comment 5* by the *Reviewer 1*.

• Line 69: molecular liquid crystals can definitely be studied quantitatively, studies of their structure at single molecule level are the ones prohibitively difficult.

Re: The sentence has been revised (pp3): ‘..., which are difficult to be directly observed and studied at single molecule level, to allow one to deduce a link between microscopic and macroscopic properties’.

• Line 72: ‘concave geometry’ is quite general, perhaps you could specify a little bit more the geometry of the particles.

Re: To clarify this, we have replaced the word “concave geometry” with “concave cavities between the spheroidal tips” in the revised manuscript (pp3).

• The notation N^* is typically used in literature to refer to a chiral nematic phase.

Re: We are sorry for the confusion. In response, we have replaced the notation “ N^* ” with “BP” in the revised manuscript and Supplementary Information.

• Line 96: Give the units for RD and RL in between brackets.

Re: Since R_D and R_L are the ratios of diameters and lengths of end blocks to central blocks, respectively, they don’t have units.

• It would be useful to note somewhere at the beginning of the paper that the particles used in this work are made out of silica.

Re: We are sorry for the confusion. We have replaced the word “DBC” with “silica DBC” when we first mentioned them in the introduction, the results, and the methods in the revised manuscript.

• Figure 4.e is not mentioned anywhere in the main text.

Re: Figure 4e is mentioned in the sentence: ‘Figure 4a,e show the configuration of a single column extracted from the simulation box.’.

• Perhaps it would help the reader to compare the experimental and simulated phase diagram in the same figure.

Re: The boundary dashed lines taken from the experimental phase diagram are used in the simulated phase diagram for comparison.

• While the section where the magnetic field manipulation shows the successful external manipulation of LCs formed by colloidal dumbbells, it feels it’s breaking the flow of the paper as it does not add a lot to its main message, which would be the phase behavior of colloidal dumbbells and formation of chiral phases.

Re: Phase switching of liquid crystals is an important property for the use of liquid crystals in such as displays and waveguides. Although we respect the reviewer’s comment, we prefer to keep this section in the manuscript to demonstrate the magnetic switchability of our colloidal liquid crystals.

Reviewer #2 (Remarks to the Author):

General comments: The authors use these particles to study the phase behavior as a function of shape (the diameters and lengths of the segments). They convincingly confirm the existence of such a phase in experiment, besides several other phases, and map out the phase diagram. These results are impressive and of high importance. They are on the whole well supported by the experimental evidence, however, I will raise a few comments below. I expect this work will serve as a benchmark for future studies, both in the molecular and colloidal field. The work is also of considerable interest for the development of switchable optical materials. I recommend acceptance of this manuscript, but think that a number of points should be elaborated first:

Re: We thank the reviewer for the positive comments.

Comment 1: The evidence (shown in Figure 2, as well as in SI) for the N-star and N1 phases is compelling. However, I found it difficult to recognize the N2 structure. There seem to be only a few isolated groups of particles arranged in the proposed structure. Could the authors provide more convincing images?

Re: In response, we have used the *in situ* confocal microscopy to further confirm the N2 structure, in addition to SEM and POM characterizations. Our confocal microscopy images of the *in situ* assembled structure of N2 phase are consistent with previous SEM and POM results. The new results have been included as a new Fig. 2g in the revised manuscript and a new Supplementary Fig. 4 in the revised Supplementary Information. Please see more details in our response to **Comment 1** by the **Reviewer 1**.

Comment 2: The particles in Figures 2j and 2k don't have a clear dumbbell shape, as R_D is very close to unity. It would be much more convincing if the authors showed an SEM image of a phase formed by rods with a large R_D (corresponding to one of the other points in Fig. 3).

Re: In response, we have added two new SEM images of two DBCs with large R_D in a new Supplementary Fig. 9 in the revised Supplementary Information.

Comment 3: I am surprised that the authors didn't mention that the colloidal dumbbells are made of silica. Not in the introduction, nor in the results, and not even in the methods does the word silica appear. This is a curious omission.

Re: We are sorry for the confusion. We have replaced the word "DBC" with "silica DBC" when we first mentioned them in the introduction, the results, and the methods in the revised manuscript.

Comment 4: As evidence of the formation of colloidal LC phases it is mentioned (on page 4) that these phase behaved as a fluid when tilted. But the authors (and readers) should be aware that even colloidal crystals (that is, colloidal solids) display such fluid behaviour as a result of their extremely low yield stress.

Re: We thank the reviewer for the insightful comments. Besides the fluidic behavior, the diffusion of silica rods on the single particle level was previously studied and used to confirm the formation of colloidal LC phases. In response, we have added a citation to this statement as a new reference 35, and revised the sentence (pp4): '...The sedimentation exhibited fluidic behavior and could flow

upon tilting the capillary tube (Supplementary Fig. 2), in accordant with the colloidal LC behavior reported before³⁵.

Comment 5: *The nematic N1 and N2 phases seem to be rather uncommon. Have they been observed before? Since nematic phases are supposed to have only orientational order and no positional order can these really be called nematics? Please discuss.*

Re: To the best of our knowledge, both N1 and N2 phase have not been observed before. According to the confocal microscopy results with different depths, the positional order only exists in two dimensions (Figure 2, Supplementary Fig. 4 and Fig. 5), rather than in three dimensions. We, therefore, prefer to call it nematics.

Comment 6: *Please discuss how the orientational order parameter is determined from SEM images.*

Re: We thank the reviewer for the constructive comment. In response to the reviewer's comment, we have used 2D confocal microscopy to measure the orientational order parameter. Please see more details in our response to **Comment 4** by the **Reviewer 1**.

Comment 7: *The authors refer to the POM images with phrases such as “the characteristic texture” or “typical schlieren texture” without further explanation. It would be helpful to interpret these images with further explanations, or provide references.*

Re: We thank the reviewer for the constructive comment. In response to the reviewer's comment, we have added citations to each texture (References 32, 36, and new 37 and 39). Please see more details in our response to **Comment 2** by the **Reviewer 1**.

Comment 8: *In Figure 4d, aren't the smectic layers tilted with respect to the director? In that case it should not be called Smectic A, but rather Smectic C.*

Re: We are sorry that the viewpoint we selected caused the false impression. In response, we have changed Figure 4d in a different viewpoint.

Figure 4 | LC phases observed in Brownian dynamics simulations. Simulation results of LC phases assembled from DBCs with (a) $R_D = 1.40$ and $R_L = 0.15$ in BP phase (right: a double-twisted column extracted from left), (b) $R_D = 1.40$ and $R_L = 0.50$ in N2 phase (right: side and bottom view of the typical local structure), (c) $R_D = 1.40$ and $R_L = 1.00$ in N1 phase (right: a cylinder picked out along the z-axis), and (d) $R_D = 1.40$ and $R_L = 1.80$ in the SmA phase (right: a cylinder picked out along the z-axis). (e) Double-twisted structure viewed from different directions.

Comment 9: In the experiment using a magnetic field to orient the LC phase, was the change reversible? This would be important to the applications that the authors mention.

Re: Yes, they are reversible. After removing the magnetic field, it usually takes a few days to go back.

Comment 10: In the Methods section I assume the particles were first transferred from ethanol to DMSO by centrifugation? I also assume that this was pure DMSO and the colloids are charge stabilized. What can the authors say about the softness of particle interactions? What was the length of the electric double layer in the solvent? Was the ionic strength measured? This is important for a better comparison with theory/simulations.

Re: We thank the reviewer for the insightful comments and suggestions. In response, we have measured the pH of the DBC dispersions in DMSO with a volume fraction of 20% to be around 9.5, corresponding to the ionic strength of 1.58×10^{-5} mol/L. Therefore, Debye Length k^{-1} can be calculated to be 76.4 nm, according to the following equation:

$$k^{-1} = \left\{ \frac{\epsilon_0 \epsilon_r k_B T}{2000 e^2 N_A} \right\}^{1/2}$$

where ϵ_0 is vacuum permittivity, ϵ_r is relative permittivity, k_B is the Boltzmann constant, T is the temperature, e is the charge of electron, I is the ionic strength, and N_A is Avogadro constant.

According to the confocal microscopy results, the distance of two adjacent DBCs is less than 50 nm, so there are repulsion interactions between them.

Comment 11: *In the Methods it is stated that “the sediment was taken out and slowly dried in air”. Wasn’t this the other way around?*

Re: Yes, we have changed to “the sediment was slowly dried in air at room temperature and taken out” in the revised manuscript.

Comment 12: *The word “freezing” in the phrase “the DBCs were constructed by freezing linearly arranged...” is unclear. Please rephrase.*

Re: We agree with the referee. In response, we have changed “freezing” to “fixing” in the revised manuscript.

Reviewer #3 (Remarks to the Author):

Key comments: *The paper by Yang Yang et al. deals with the formation of liquid crystalline phase from dumbbell-shaped colloids. Specifically, one of their claims is the evidence of the chiral organization reminiscent of double twist cylinders observed in blue phases from their achiral particles. Note that for steric reason, it is expected to get twist between the particles so the result is not for me highly surprising. Interestingly, the authors obtain also other mesophases by tuning the geometrical features of their particles, including two nematic phases (one-lock (N1) and two-lock (N2) structures) and one smectic phase. However, the experimental proofs of such claims is rather poor for a publication in a journal like Nature Communications: beyond low magnification polarizing microscopy (which only gives indirect structural information and nothing at the single particle level), only scanning electron microscopy has been performed to give insights about the liquid crystalline organization at the particle scale. However such an electron microscopy technique is intrinsically obtained on dried samples. Direct observations of particles in suspensions, which means at equilibrium, seem feasible, as shown both by the micrometer size of their particles and by their possible labeling by FITC dyes (as indicated in Supplementary Figure 2). Therefore I strongly recommend to the authors to partially label some of dumbbell-shaped colloids and to perform extensive study of their suspensions by optical confocal microscopy in order to get direct structural demonstration of their liquid crystalline phase at equilibrium, as well as some information of the dynamics of the system.*

Re: We thank the reviewer for the constructive suggestion and insightful comments. As suggested by the reviewers, we labelled the DBCs with FITC and used 3D confocal microscopy to characterize different LC phases *in situ*. We agree with the reviewer that the experimental data is not sufficient to clearly present the LC phases in our original submission. We have performed additional confocal microscopy experiments and made substantial changes in the revised manuscript. Please see our detailed response to the comments as follows.

Comment 1: *The absence of overall chirality in the suspensions should be proved by providing proper statistics (See line 109 p.5), and not only a claim based on one (or a few) image(s).*

Re: As the reviewer suggested, we have used 3D confocal microscopy to characterize the BP phase as shown in the new Fig. 2, Supplementary Fig. 3, Supplementary Video 1 and 2. Our statistical analysis is based on over 100 double-twisted columns in the 3D confocal microscopy images of each sample. Please see more details in our response to **Comment 1** and **Comment 3** by the **Reviewer 1**.

Comment 2: *Still dealing with chirality, I guess that the unit of the helical pitch in Figure 3b is in μm (and not in nm as it is written).*

Re: We are sorry for the confusion. The unit in Figure 3b was in micrometers. We have corrected the units in Figure 3b.

Comment 3: *Why the authors have not varied the initial volume fraction of their colloids to prepare their samples? Mostly the geometric features of their dumbbells have been changed (See Fig. 3a) but it would be as important to get the dependence of the phase behavior as a function of the particle volume fraction (for a given size).*

Re: We thank the reviewer for the helpful comments and suggestions. In response, we have carried out additional experiments on the DBC sample with five different initial volume fractions, including 10, 15, 20, 25%, and 30%, but the BPs were formed for all these initial volume fractions we studied. This is probably due to the similar volume fraction of the sediment phase after ~30 days of sedimentation. Different sedimentation time or smaller initial volume fractions should lead to the large change of the volume fraction in a wide range of the sediment phase. The studies are currently ongoing in our laboratory, and the research will be reported in a separated manuscript in the future.

Comment 4: *Some claims in the abstract and in the introduction are clearly overstated. For instance, it is mentioned the blue phase based LCDs which are indeed interesting for their fast response. However, mentioning LCD based on colloidal particles is certainly awkward, considering the very slow response time of colloids, the difficulty of applying electric fields in colloidal suspensions, etc...*

Re: We thank the reviewer for the helpful comment. In response to the reviewer's comment, we have revised the abstract and introduction. Please see more details in our response to **Comment 5** by the **Reviewer 1**.

** See Nature Research's author and referees' website at www.nature.com/authors for information about policies, services and author benefits.

COVID 19 and impact on peer review

As a result of the significant disruption that is being caused by the COVID-19 pandemic we are very aware that many researchers will have difficulty in meeting the timelines associated with our peer

review process during normal times. Please do let us know if you need additional time. Our systems will continue to remind you of the original timelines but we intend to be highly flexible at this time. This email has been sent through the Springer Nature Tracking System NY-610A-NPG&MTS

Confidentiality Statement:

This e-mail is confidential and subject to copyright. Any unauthorised use or disclosure of its contents is prohibited. If you have received this email in error please notify our Manuscript Tracking System Helpdesk team at <http://platformsupport.nature.com>.

Details of the confidentiality and pre-publicity policy may be found here <http://www.nature.com/authors/policies/confidentiality.html>

Privacy Policy | Update Profile

REVIEWER COMMENTS

Reviewer #1 (Remarks to the Author):

After the peer-review process of this work, the authors have improved the quality of the experimental data. In particular, the addition of the confocal data strengthens the experimental evidence for the observation of Chiral Blue Phase, which was one of my main concerns in the previous version of the work. However, I still have concerns about the soundness of this work.

My concerns are regarding the discussion of the experimental results and the general pitch of the paper. In other words, I have still found too many technical and conceptual inaccuracies to consider this work to be accepted for publication in Nature Communications. In my opinion, the following points are still not clear:

1. In general, the authors seem to put a lot of emphasis on the finding that the geometry of the colloids affect their phase behaviour. However, this is not a novel finding as it was first discovered by Onsager in 1949, and it has been extensively reported since then in many works, specially in the liquid crystal community. Instead, I think the authors need to emphasise that (i) they find the key geometrical parameters controlling the DBCs phase behaviour (i.e. RD and RL), and (ii) that for a given range (i.e. $RL < 0.45$ and $RD > 1.1$) they find the first experimental observation of a colloidal chiral blue phase.
2. The authors have extended the introduction, which now described what a Chiral Blue Phase is. While I appreciate the authors have addressed this point, they describe the distinction between three different types (BP I, II and III). However, this classification is not mentioned again in the discussion, i.e. they do not discuss which type is the phase they find.
3. Also, in the last part of the second paragraph in the introduction, when they say, 'Colloidal LCs formed by DBCs would enable the visualization of the twisted structure at the single particle level, thus providing new insights into how chirality can emerge from achiral building blocks.' it is not clear if with 'the twisted structure' they refer to the chiral blue phase. I believe they are, because in such case, the 'gap' of knowledge presented in the introduction (i.e. the missing observation of the blue phase) would be clear. Unfortunately, this key point is not clearly written or discussed, and therefore is misleading.
4. In the following sentence of paragraph 2 in the introduction, the authors state that 'to date, the manner in which the concave geometry of DBCs determines their LC phase behaviour has not been experimentally explored'. However, in the same paragraph, but two sentences above they discuss how

the aspect ratio of the colloids of another system (ref 23), determines a net-like, zipper-like, and weave-like structures, i.e. that the geometry of the colloids affect their phase behaviour.

5. In the results section (LC phases assembled from DBCs), the sentence 'Statistically, the population of right- and left-handed chiral columns is approximately equal at a 100 μm length scale', is just not clear. What the relevance of a length scale when comparing if the populations of left- and right-handed columns are similar or different? This sentence needs elaboration.

6. The authors also mention that the handedness is measured from confocal images (such as those shown in Figure S3), however, there is not enough experimental evidence as to how the authors measure that. Are they using image analysis techniques from confocal images (this might prove to be difficult as the particles in the images shown in Fig S3 are very close together)? Or eye measurements? In my opinion, SEM images in Fig 1 seem to be the clearest images providing such information.

7. The drawings of the particles on top of the SEM images in Figure 1 continue to make the interpretation misleading. As mentioned in my previous report, the 'dotted particles' in Figure 2F do not match with physical particles, which I find very misleading.

8. The authors measure the 2D order parameter from confocal images and describe the formula that they use to do it. However, they don't explain how they actually calculate it; i.e. do they measure manually the orientation of each particle, do they use image analysis techniques?

Some other minor points:

- The sentence 'namely, the ratio of diameters (R_D) of end blocks (D_e) to central blocks (D_c), and the ratio of lengths (R_L) of end blocks (L_e) to central blocks (L_c)' is rather confusing. I would re-phrase it as follows: 'namely, the ratio of diameters of end blocks to central blocks (R_D), and the ratio of lengths of end blocks to central blocks (R_L), see schematic Figure 1'.
- The caption of Figure 1 doesn't explain all the information contained in the Figure, such as what R_e , D_e , P , t_1 , t_2 stands for.
- In LC phases assembled from DBCs section: replace 'the sedimentation exhibited' by 'the sediment exhibited...'
- Check spelling: i.e. 'in accordance' instead of 'in accordant'..,

- The title of the sections is not really correct: Instead of Results it would probably more appropriate to call it Results and Discussion, and instead of Discussion, replace it by Conclusions and Outlook.
- The title could also be simplified by 'Liquid-crystalline behaviour on dumbbell-shaped colloids and the observation of chiral blue phases'.
- In the Supplementary Figure 3, state the z-position of each snapshot.

Reviewer #2 (Remarks to the Author):

This manuscript describes observations of the phase behaviour of colloids in the shape of dumbbells. Several new liquid crystalline phases are reported, among which is a chiral phase. In addition to data already included in the first version of this manuscript the authors added confocal microscopy observations to show the identity of the phases without having to dry them first. Their phase diagram represents an impressive amount of experimental work in which the geometrical parameters of the particles are varied. It is also in good agreement with their simulations. These results are convincing and important in the drive to understand the relation between shape and LC formation, in particular how a chiral phase is formed in a system of achiral building blocks.

My concerns to the previous version of the manuscript have all been addressed. One small omission should still be corrected: In the phase manipulation experiments with magnetic fields it is not mentioned which particles were used. To sum up: I recommend acceptance.

REVIEWER COMMENTS

Reviewer #1 (Remarks to the Author):

General comments: After the peer-review process of this work, the authors have improved the quality of the experimental data. In particular, the addition of the confocal data strengthens the experimental evidence for the observation of Chiral Blue Phase, which was one of my main concerns in the previous version of the work. However, I still have concerns about the soundness of this work. My concerns are regarding the discussion of the experimental results and the general pitch of the paper. In other words, I have still found too many technical and conceptual inaccuracies to consider this work to be accepted for publication in *Nature Communications*.

Re: We thank the reviewer for the insightful and valuable comments. We agree with the reviewer that some results were not clearly written or discussed in our last submission. In response, we have made substantial changes to make the discussion accurate in the revised manuscript. Please see our detailed response to the comments as follows.

Comment 1: In general, the authors seem to put a lot of emphasis on the finding that the geometry of the colloids affect their phase behaviour. However, this is not a novel finding as it was first discovered by Onsager in 1949, and it has been extensively reported since then in many works, specially in the liquid crystal community. Instead, I think the authors need to emphasise that (i) they find the key geometrical parameters controlling the DBCs phase behaviour (i.e. R_D and R_L), and (ii) that for a given range (i.e. $R_L < 0.45$ and $R_D > 1.1$) they find the first experimental observation of a colloidal chiral blue phase.

Re: We very appreciate the reviewer's invaluable comments and suggestions. As required by the reviewer, we have revised the corresponding parts in the abstract, introduction and conclusion. Accordingly, we have added the following statements in the revised manuscript.

(pp2, line33): '..., and particularly the first experimental observation of BP III with double-twisted chiral columns.'

(pp4, line86): '...The phase behaviours of DBCs are found to be dictated by the two key geometrical parameters of DBCs, namely, the ratio of diameters of end blocks to central blocks (R_D), and the ratio of lengths of end blocks to central blocks (R_L), see schematic Fig. 1. Most importantly, this work demonstrate the first experimental observation of colloidal chiral BP III. In particular, the colloidal chiral BP III was formed from achiral building blocks in the range of $R_D > 1.1$ and $R_L < 0.45$.'

(pp8, line216): '...Both experimental and simulation results confirmed that R_D and R_L are critical for the formation of mesoscopically ordered phases.'

(pp8, line220): '...It is worth noting that double-twisted chiral superstructures with racemic mixtures, namely BP III, can be produced from rigid achiral colloids in the range of $R_D > 1.1$ and $R_L < 0.45$.'

Comment 2: The authors have extended the introduction, which now described what a Chiral Blue Phase is. While I appreciate the authors have addressed this point, they describe the distinction between three different types (BP I, II and III). However, this classification is not mentioned again

in the discussion, i.e. they do not discuss which type is the phase they find.

Re: We thank the reviewer for the constructive comment. The classification of BP III has been added in the discussion section in the revised manuscript (pp4, line108) as follows: ‘...Since the double-twisted chiral columns are randomly oriented and no ordered packing is found, the formed phase of DBCs is BP III phase with a racemic mixture of chiral columns.’.

We also used ‘BP III’ to replace ‘BP’ in the rest of revised manuscript.

Comment 3: *Also, in the last part of the second paragraph in the introduction, when they say, ‘Colloidal LCs formed by DBCs would enable the visualization of the twisted structure at the single particle level, thus providing new insights into how chirality can emerge from achiral building blocks.’ it is not clear if with ‘the twisted structure’ they refer to the chiral blue phase. I believe they are, because in such case, the ‘gap’ of knowledge presented in the introduction (i.e. the missing observation of the blue phase) would be clear. Unfortunately, this key point is not clearly written or discussed, and therefore is misleading.*

Re: We are sorry for the misleading. Accordingly, the following statements have been revised in the revised manuscript (pp3, line69): ‘...Therefore, great efforts are needed to develop colloidal BP LCs for the direct observation of BP.’.

(pp3, line78): ‘...Colloidal LCs formed by DBCs would enable the visualization of the chiral/blue phases with twisted/double-twisted structures at the single particle level,....’.

Comment 4: *In the following sentence of paragraph 2 in the introduction, the authors state that ‘to date, the manner in which the concave geometry of DBCs determines their LC phase behaviour has not been experimentally explored’. However, in the same paragraph, but two sentences above they discuss how the aspect ratio of the colloids of another system (ref 23), determines a net-like, zipper-like, and weave-like structures, i.e. that the geometry of the colloids affect their phase behaviour.*

Re: We apologize for the confusion. The assembly of DBCs in crystal state has been studied, but the LC phase behaviour of DBCs has not been studied. Accordingly, we have added the following statements in the revised manuscript (pp3, line74): ‘...Dumbbell-like nano-arrows can assemble into net-like, zipper-like, and weave-like two-dimensional lattices, as well as non-close-packed three-dimensional supercrystals but not in LC state, depending on the aspect ratio of the colloids²⁵.’.

Comment 5: *In the results section (LC phases assembled from DBCs), the sentence ‘Statistically, the population of right- and left-handed chiral columns is approximately equal at a 100 μm length scale’, is just not clear. What the relevance of a length scale when comparing if the populations of left- and right-handed columns are similar or different? This sentence needs elaboration.*

Re: We gratefully appreciate the reviewer’s comment. We have counted 100 columns, which are distributed at around a 100 μm length scale, for statistical purposes. In response, we have revised the statement, as read (pp5, line116): ‘...Statistically, the populations of right- and left-handed chiral columns are approximately equal by counting 100 columns.’.

Comment 6: *The authors also mention that the handedness is measured from confocal images (such as those shown in Figure S3), however, there is not enough experimental evidence as to how the authors measure that. Are they using image analysis techniques from confocal images (this might prove to be difficult as the particles in the images shown in Fig S3 are very close together)? Or eye*

measurements? In my opinion, SEM images in Fig 1 seem to be the clearest images providing such information.

Re: We are sorry for the confusion. The handedness of the double-twisted column was determined by eye measurements of both SEM (Fig. 2a,b) and 3D confocal images (Supplementary Video 1 and 2). Though the 2D confocal images (Fig. 2c) are not as clear as the SEM images (Fig. 2a,b), the rotation direction of columns, i.e., the handedness of the columns, can be well defined from the 3D confocal images with different depths, as shown in Supplementary Video 1 and 2. In response, the statement has been revised, as read (pp5, line113): ‘...Though the 2D confocal images (Fig. 2c) are not as clear as the SEM images (Fig. 2a,b), the rotation direction of columns, i.e., the handedness of the columns, can be well defined by eye measurement of 3D confocal images with different depths (as shown in Supplementary Video 1 and 2).’.

Comment 7: The drawings of the particles on top of the SEM images in Figure 1 continue to make the interpretation misleading. As mentioned in my previous report, the ‘dotted particles’ in Figure 2F do not match with physical particles, which I find very misleading.

Re: We sorry for the misleading and agree with the reviewer. In response, we have removed the dotted lines on top of the particles in Figure 2f, g, j, and k in the revised manuscript.

Comment 8: The authors measure the 2D order parameter from confocal images and describe the formula that they use to do it. However, they don’t explain how they actually calculate it; i.e. do they measure manually the orientation of each particle, do they use image analysis techniques?

Re: We are sorry for the confusion. In response, we have revised the Methods. Accordingly, we have added the following statement in the revised manuscript (pp10, line264): ‘...We have manually measured the angles between the director and the long axis of each DBC for 200 DBCs of each sample using Image J software that is available at NIH Web site, to calculate S_{2d} for N1 and N2 phase, respectively.’.

Some other minor points:

- The sentence ‘namely, the ratio of diameters (R_D) of end blocks (D_e) to central blocks (D_c), and the ratio of lengths (R_L) of end blocks (L_e) to central blocks (L_c)’ is rather confusing. I would rephrase it as follows: ‘namely, the ratio of diameters of end blocks to central blocks (R_D), and the ratio of lengths of end blocks to central blocks (R_L), see schematic Figure 1’.

Re: We thank the reviewer for the very helpful suggestion. In response, the sentence has been revised according to the reviewer’s suggestion, as read (pp4, line86): ‘...The phase behaviours of DBCs are found to be dictated by the two key geometrical parameters of DBCs, namely, the ratio of diameters of end blocks to central blocks (R_D), and the ratio of lengths of end blocks to central blocks (R_L), see schematic Fig. 1.’.

- The caption of Figure 1 doesn’t explain all the information contained in the Figure, such as what R_e , D_e , P , t_1 , t_2 stands for.

Re: In response, we have added the following statement in the caption of Figure 1 (pp16, line413): ‘... D_e and D_c are the diameters of end block and central block, respectively. L_e and L_c are the lengths of end block and central block, respectively. R_L is the ratio of lengths of end blocks to central blocks. t_1 and t_2 are the long axis of the chiral column and the layer normal of the barrel-like

chiral twist, respectively. $1/2P$ is the half helical pitch.’.

- In LC phases assembled from DBCs section: replace ‘the sedimentation exhibited’ by ‘the sediment exhibited...’

Re: We have replaced ‘the sedimentation exhibited’ by ‘the sediment exhibited’.

- Check spelling: i.e. ‘in accordance’ instead of ‘in accordant’..,

Re: We have changed ‘in accordant’ to ‘in accordance’, and also carefully checked the spelling in the revised manuscript.

- The title of the sections is not really correct: Instead of Results it would probably more appropriate to call it Results and Discussion, and instead of Discussion, replace it by Conclusions and Outlook.

Re: We are sorry for the confusion. In response, we have replaced the title of the sections “Results” with “Results and Discussion” and “Discussion” with “Conclusions” in the revised manuscript.

- The title could also be simplified by ‘Liquid-crystalline behaviour on dumbbell-shaped colloids and the observation of chiral blue phases’.

Re: We thank the reviewer for the helpful suggestion. Accordingly, we have changed the title to ‘Liquid-crystalline behaviour on dumbbell-shaped colloids and the observation of chiral blue phases’ in the revised manuscript.

- In the Supplementary Figure 3, state the z-position of each snapshot.

Re: We have stated the z-position in the Supplementary Figure 3 in the revised Supplementary Information.

Reviewer #2 (Remarks to the Author):

General comments: This manuscript describes observations of the phase behaviour of colloids in the shape of dumbbells. Several new liquid crystalline phases are reported, among which is a chiral phase. In addition to data already included in the first version of this manuscript the authors added confocal microscopy observations to show the identity of the phases without having to dry them first. Their phase diagram represents an impressive amount of experimental work in which the geometrical parameters of the particles are varied. It is also in good agreement with their simulations. These results are convincing and important in the drive to understand the relation between shape and LC formation, in particular how a chiral phase is formed in a system of achiral building blocks. My concerns to the previous version of the manuscript have all been addressed. One small omission should still be corrected: In the phase manipulation experiments with magnetic fields it is not mentioned which particles were used. To sum up: I recommend acceptance.

Re: We thank the reviewer for the positive comments.

In response, we have added the following statement in the revised manuscript (pp8, line207):
'...Upon applying a magnetic field of 5 T, DBCs with two very short e-blocks ($R_D > 1.1$ and $R_L < 0.45$) were rotated and aligned their long axis along the field direction,...'.

And also added the following statement in the caption of Supplementary Fig. 14, as read: '...POM images of phases formed from DBCs with $L_e = 160$ nm, $L_c = 1660$ nm, $D_e = 315$ nm, and $D_c = 240$ nm...'.

.

REVIEWERS' COMMENTS

No comments to the authors